# Novel cervical pedicle screw design to enhance the safety insertion without compromising biomechanical strength

Pilan Jaipanya[1,2], Pongsthorn Chanplakorn[1]*, Annop Sikongkaew[1], Anak Khantachawan[3], Suphaneewan Jaovisidha[4], Thamrong Lertudomphonwanit[1]

1 Department of Orthopaedics, Faculty of Medicine, Ramathibodi Hospital, Mahidol University, Bangkok, Thailand, 2 Chakri Naruebodindra Medical Institute, Faculty of Medicine, Ramathibodi Hospital, Mahidol University, Samut Prakan, Thailand, 3 Department of Mechanical Engineering, Faculty of Engineering, King Mongkut's University of Technology Thonburi, Bangkok, Thailand, 4 Department of Diagnostic and Therapeutic Radiology, Faculty of Medicine Ramathibodi Hospital, Mahidol University, Bangkok, Thailand

* Pongsthornc@yahoo.com

## Abstract

### Background

Lateral mass screw (LMS) is a more widely adopted method for posterior cervical spine fixation than the cervical pedicle screw (CPS). Despite its lower pullout strength, the insertions of LMS are more reproducible and have a lower risk. CPS insertion is a technically demanding procedure due to the small pedicle channel. Thus, CPS insertion has a high risk of pedicle wall perforation, resulting in neurovascular injury. For these reasons, surgeons may avoid CPS insertion despite its benefit of greater biomechanical strength. Therefore, an improvement in the CPS design is needed to avoid this catastrophic complication.

### Objectives

To develop a new design of CPS, aiming to decrease pedicle wall perforation, while maintaining the biomechanical properties comparable to those of standard CPS.

### Materials & methods

To reduce the risk of pedicle wall perforation, a novel CPS design should be configured in tapered shape, with a tapering screw pitch and thread diameter with a self-tapping thread. A bilayer bone finite element model representing the cortical and cancellous bone of the cervical spine pedicle was used for pullout strength test. According to our CT-based study of cervical pedicle anatomy in a normal population, the final CPS was created according to the parameters that yielded the best biomechanical strength according to finite element studies. The safety of CPS insertion, in terms of pedicle wall penetration, was assessed in 3D-printed cervical spine models

**Data availability statement:** All relevant data are within the manuscript and its Supporting Information files.

**Funding:** The funding for this study is supported by the Faculty of Medicine, Ramathibodi Hospital, Mahidol University. The funders had no role in study design, data collection and analysis, decision to publish, or preparation of the manuscript.

**Competing interests:** The authors have declared that no competing interests exist.

**Abbreviations:** LMS, lateral mass screw; CPS, cervical pedicle screw; CT, computed tomography.

of C3-C7. The pullout test was subsequently performed in a tri-layer sawbones foam model to compare the novel CPS, convention CPS, and lateral mass screw.

## Results

The final screw design was a taper configuration with core diameter from 2.5 to 2.0 mm, thread diameter from 4.0 to 2.5 mm and pitch length from 1.0 to 1.25 mm. A total of 60 screws (30 conventional CPS screw and 30 Novel CPS screw) were tested in 6 3D cervical spine models. No case of pedicle wall perforation were found in the novel-design CPS group. In the conventional CPS group, 8 pedicle wall perforations were encountered, which was a statistically significant difference (p = 0.002). The novel CPS screw design and conventional CPS screw yielded pullout strengths of 449.7 N and 495.0 N, respectively, which showed no statistical difference. The LMS screw yielded a pullout strength of 168.3 N, showing statistically less strength compared with the 2 types of CPS screws.

## Conclusions

The proposed novel CPS could decrease pedicle wall perforation and enhance the safety of screw insertion. Its pullout strength is comparable to that of a 3.5-mm standard CPS and superior to that of a 3.5-mm lateral mass screw.

## Introduction

Cervical pedicle screw (CPS) fixation has emerged as a promising technique in cervical spine surgery. Despite the lack of guideline on the type posterior cervical fixation, CPS offers multiple benefits over other forms of fixation, including improved stability and better load-sharing capacity [1]. CPS fixation is particularly useful in many cervical spine surgeries where a strong and stable fixation is crucial to mitigate the risk of non-union and pseudarthrosis, allowing for a bony solid fusion [2,3]. However, despite its advantages, the cervical pedicle channel is relatively small and has a complex three-dimensional anatomy, making accurate screw placement a demanding task [4]. Inaccurate screw placement can result in detrimental complications such as neurovascular injury, spinal cord compression, and screw loosening [5]. Through our knowledge, there are no specific guidelines or recommendation for pedicle screw design. The most common screw configuration for CPS fixation is a cylindrical shaped design with 3.5 mm diameter.

Considering the small internal diameter of cervical pedicles, as reported by Chanplakorn et al., the cervical pedicle screws should be redesigned to reduce complications while maintaining their biomechanical strength [6]. Modifying the screw thread design, such as the pitch, depth, or profile, in combination with the conical shape, can potentially reduce pedicle wall perforation. This could also help maintain the fixation strength, while reducing complications such as screw loosening or pullout. Our novel CPS design proposes a tapered, conical-shaped screw with a slightly smaller

tip diameter which would allow for facilitating insertion in the varieties of cervical pedicle dimensions amongst patients. Furthermore, this screw configuration would allow for a more secure fit within the pedicles and reduces the risk of cortical breaches and neurovascular injuries.

Currently, there is limited literature specifically considering conical-shared screws for CPS fixation. Nevertheless, conical screws have been shown to provide better pullout strength compared to cylindrical screws in lumbar pedicle screw fixation, particularly in osteoporotic bone [7,8]. This study aimed to develop a new design of CPS, allowing for safer instrumentation with comparable biomechanical strength when compared with conventional CPS.

## Materials and methods

This study has been approved by the institution's research committee (MURA2020/115). The study was divided into three phases, including

1. Screw design with finite element test

2. The safety insertion test in the 3D reconstruction model

3. The pullout strength test.

The protocol for using CT scan data to create 3D models was approved by the institutional IRB as a minimal risk procedure. Since this study did not involve human participants or specimens, the requirement for informed consent was waived.

### Screw design with finite element test

The screw design was based on the cervical pedicle anatomical study from CT scan-based analysis by Chanplakorn et al. [6]. Findings from this previous study revealed that the cervical pedicle has a thick inner cortex and a small intramedullary canal [6]. Thus, the conventional cylindrical screw design with a standard 3.5 mm diameter has a limitation to insert into the small inner canal. A novel cervical pedicle screw design involving a tapering core and thread diameter would lessen the chance of pedicle wall penetration (Fig 1). The details of screw parameters are presented in Table 1. However, a tapering screw design would significantly reduce the pullout strength when compared with the conventional screw design. Thus, we propose that increasing the thread diameter of the upper part of the screw would significantly increase the overall pullout strength.

A bilayer bone finite element model representing the cortical and cancellous bone of the cervical spine pedicle was created using Simsolid software for structural analysis [INDIELEC, Parque Tecnológico de Valencia] for the testing of screw biomechanical properties. The young modulus of cortical and cancellous bone was 1.24 Pa and 1.0 Pa, respectively. The Poisson's ratio for cortical and cancellous bone ratio was 0.3 and 0.25, respectively. A finite element analysis comparing the pullout strength of the novel cervical pedicle and the conventional 3.5 mm x 22 mm pedicle screws was conducted. The design of the tapering configuration screw was then readjusted to maximize its pullout strength, as shown in Table 1.

Then, the parameters of the new cervical screw model 2 were used to manufacture the titanium alloy screws, as this model yielded the maximal pullout strength (Fig 1) to use for the remained experiments.

### The safety of insertion test

3D printed models were created to test the safety of insertion of the conventional and novel cervical pedicle screw design. The conventional screw used was a multiaxial cervical pedicle screw, 3.5 mm diameter, 22 mm length (VuePoint OCT, NuVasive, San Diago, USA).The novel pedicle screw was manufactured from titanium alloy using the final model from phase 1, yielding the maximal pullout strength.

The 3D-printed models were created from CT scan of the normal C3-C7 cervical spine. The lamina and spinous processes were subtracted from raw CT image to better visualize the pedicles. The pedicle aiming tracks were created by a

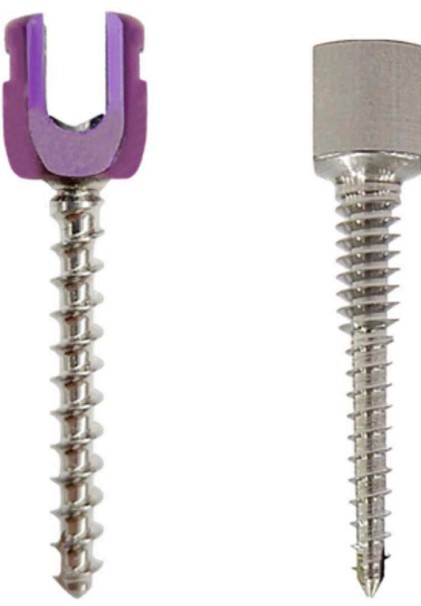

**Fig 1. A novel cervical pedicle screw design (right) involving a tapering core and thread diameter would lessen the chance of pedicle wall penetration when compared with the conventional cervical pedicle screw (left).**

**Table 1. Parameters of screws.**

| Parameters | Standard CPS | New CPS Model 1 | New CPS Model 2 |
|---|---|---|---|
| Material | Ti-6AI-4V ELI | Ti-6AI-4V ELI | Ti-6AI-4V ELI |
| Design | Cylindrical | Conical | Conical |
| Thread diameter | 3.5 mm | 4.0 to 2.5 mm | 4.0 to 2.5 mm |
| Core diameter | 2.0 mm | 2.5 to 2.0 mm | 2.5 to 2.0 mm |
| Pitch | 2.0 mm | 1.25 mm | 1.00 to 1.25 mm |
| Normal stress (Mpa) | −0.91 to 3.11 | −8.32 to 3.14 | −9.69 to 3.41 |
| Pullout strength (Mpa) | 6.57 | 6.41 | 7.16 |

line drawn from the posterior bony surface to the middle of the pedicle canal. The aiming tract was 1.5 mm in diameter, extending from the dorsal bony surface to just above the cervical pedicle canal (Fig 2). Each aiming tract was created in a CT scan image for each pedicle individually to ensure the optimal screw insertion trajectory before model printing. This process was performed using image processing software. The final testing models were printed by a 3D printer using Acrylonitrile butadiene styrene (ABS). The model was loaded into the special design immobilizer to secure its position throughout the screw insertion experiment (Fig 3).

In the screw insertion safety test, the type of screw inserted was determined using concealed paired envelopes in a randomized manner. The first envelope determines the cervical level and side for initial screw insertion. The second envelope determines the type of screw (i.e., conventional versus novel screw design). For each cervical level, the contralateral side pedicle was inserted with the counterpart screw type. A single senior orthopedic spine surgeon inserted all screws according to the screw insertion technique provided by the manufacturer. The pedicle aiming track was used to navigate the predrill procedure for all screw insertions.

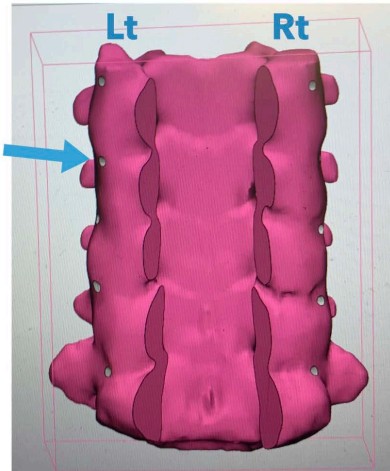 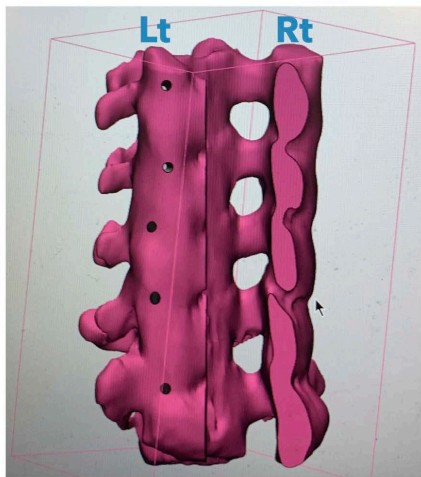

**Fig 2. The 3D model of C3-C7 cervical spine with subtraction of lamina and spinous processes with screw aiming tract (blue arrow) in an image processing software before 3D printing.**

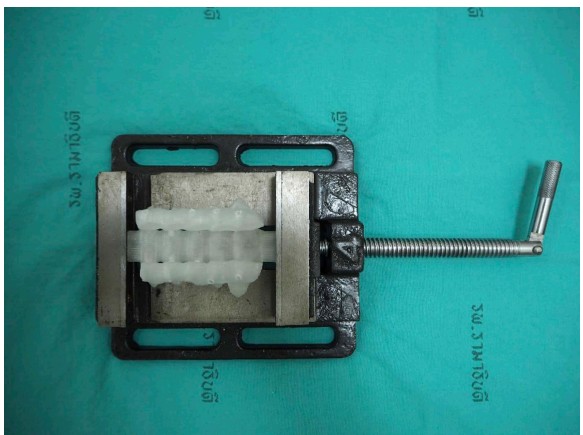

**Fig 3. The 3D printed model (white) was loaded into the specially designed immobilizers to secure their positions throughout the screw insertion experiment.**

## Screw insertion technique

1. Conventional cervical screw insertion technique. The predrilling was performed using a 2.05-mm drill, followed by a 3.5-mm tap before screw insertion. This technique was applied for cervical pedicle screw and lateral mass screw insertion.

2. The novel cervical screw insertion technique. Predrilling was performed using a 2.00-mm drill, followed by direct screw insertion without tapping, as the novel screw design incorporated a self-tapping tip. The slightly smaller drill diameter compared to the 2.05-mm drill used for conventional screws was selected to optimize thread engagement in the absence of tapping. The 0.05-mm difference is within practical tolerance and unlikely to affect mechanical outcomes significantly in polyurethane foam models. The primary biomechanical distinction between techniques lies in the

absence of tapping, which alters the screw–bone interface characteristics and may contribute more substantially to pullout performance than the minor variation in pilot hole diameter.

Pedicle wall perforation was evaluated by direct visualization. Pedicle wall perforation was classified using Neo classification: Grade 0, no deviation, and the screw was contained in the pedicle. Grade 1, deviation less than 2 mm (i.e., less than half of the screw diameter). Grade 2, deviation more than 2 mm and less than 4 mm. Grade 3, deviation more than 4 mm (i.e., complete deviation) [9].

## The pullout strength test

The pullout strength test was conducted on solid rigid polyurethane foam, adhering to the American Society of Testing Materials (ASTM F1839; ASTM, USA) standard for orthopedic devices and instrument testing. The properties of the tested foam material were representative of the density of cortical bone (0.32 g/cc [20 PCF]) and cancellous bone (0.24 g/cc [15 PCF]). Two types of models were created: 1. The tri-layer foam model (cancellous-cortical-cancellous bone), which was designed to represent the longitudinal axis of the pedicle screw insertion, the lateral mass, pedicle, and vertebral body. The tri-layer foam was made from 2 layers of cancellous bone on the first and third layer with cortical bone in the middle layer; the thickness of each layer was 10 mm. All layers of foam were attached with glue and then cut into 30 mm × 15 mm × 30 mm dimensions, 2. The mono-layer cancellous bone foam model represents the lateral mass for lateral mass screw insertion. They were prepared into 30 mm × 15 mm × 30 mm dimensions, as shown in Fig 4.

In the tri-layer foam model, representing the anatomy for cervical pedicle screw insertion, five screws each of 3.5 mm x 22 mm conventional cervical pedicle screw and the novel design cervical pedicle screw were inserted. In the mono-layer foam model, representing the anatomy for lateral mass screw insertion, multi-axial 3.5 x 16 mm screws (VuePoint OCT, NuVasive, San Diego, USA) were inserted. No LMS insertion was performed using the novel CPS, as its design and dimensions were not intended for lateral mass trajectories. Each screw was then connected to the testing machine using a special jig (Fig 5). The foam model with a screw was then placed in such a manner that the pedicle-screw axis was co-axial with the pullout direction on the jig. A 5 mm/min of the tensile load was applied to the test specimen until the screw was released from the test block. Load and displacement values were recorded, and the maximum load generated during screw pullout was defined as the pullout strength. While the conventional screw was polyaxial and the novel screw was monoaxial, the pullout force was applied directly along the axis of the screw shaft using a custom fixture. Therefore, the screw head articulation (polyaxial vs. monoaxial) was not engaged during testing and is not expected to influence the measured pullout strength.

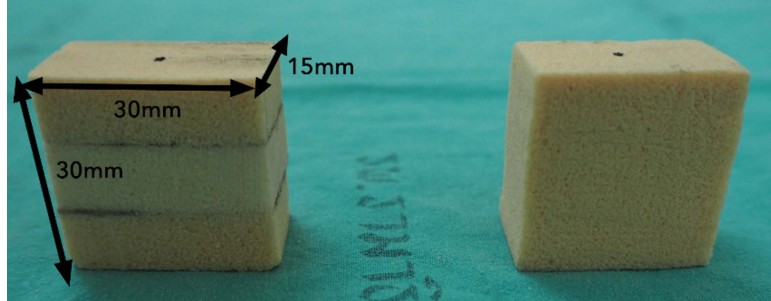

**Fig 4. The Tri-layers foam (left) and the mono-layer foam (right) were prepared into 30 mm × 15 mm × 30 mm dimensions.**

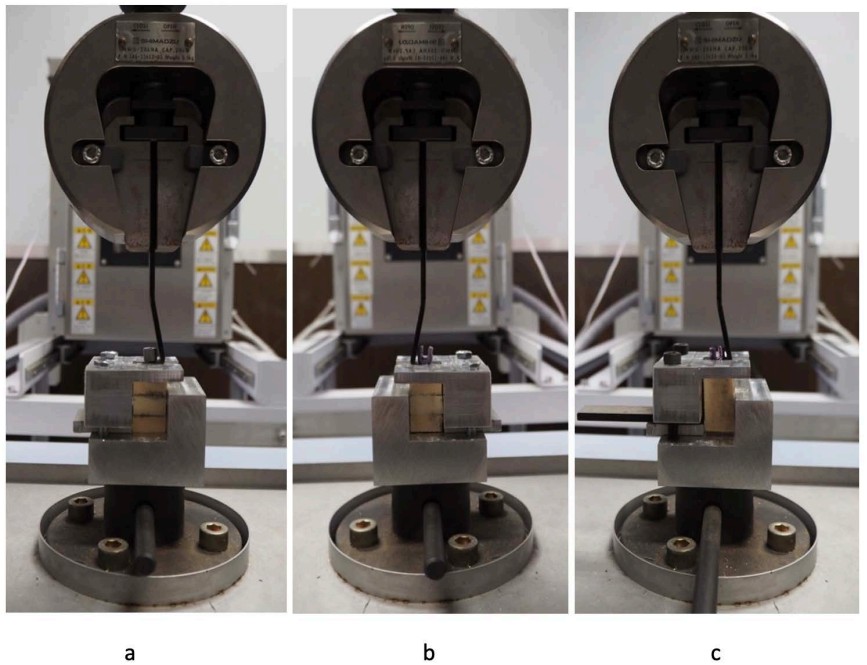

a　　　　　　　　　　b　　　　　　　　　　c

**Fig 5. A screw placement in a foam model, the new CPS screw (a), the stand CPS screw (c), and the lateral mass screw (c) were loaded in a special jig prepared for the actual pullout test.**

## Statistical analysis

Statistical analysis was performed using STATA SE version 17 (StataCorp, College Station, Texas, USA). The Fisher's exact test was used to analyze for differences in pedicle wall perforation between the new and standard cervical pedicle screw. The Bonferroni test was used to test for the pullout strength between groups. The $p < 0.05$ was considered as a statistical significance.

## Results

### 1. Screw design and finite element test

The result of the simulation from the Simsolid software is illustrated in Table 1. The pullout strength of the new cervical pedicle screw (CPS) designs was comparable with conventional screw design. The new CPS design model 2 revealed the highest pullout strength. Then, the new CPS was crafted using the following parameters: tapered conical shape core diameter, 2.5 mm at the upper part and 2.0 mm at the screw tip, thread diameter 4.0 mm with 1.0 mm pitch length at the upper 1/3 of screw and thread diameter 2.5 mm with 1.25 mm pitch length at the remaining 2/3 of screw (Fig 1, right).

### 2. The safety insertion test

The novel CPS screws and the conventional CPS screws were inserted into the 3D printed models of the cervical spine form C3-C7 for safety analysis. The CT scans of the cervical spine from 3 patients were used to create the models. One set of CT scan images was used to print 2 models. A total of 6 models, which consisted of 60 pedicles, were used in the study. A total of 60 screws were tested (30 novel design CPS and 30 standard CPS) with randomized selection order. No pedicle wall perforation was observed in the new CPS screw insertion. In contrast, Pedicle wall perforation was observed in 8 pedicles, grade 1 in 6 pedicles, and grade 2 in 2 pedicles in the conventional CPS screw insertion (Fig 6). The most

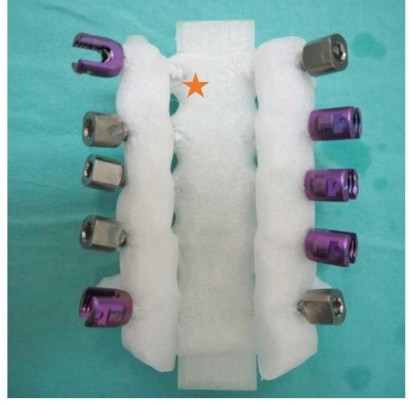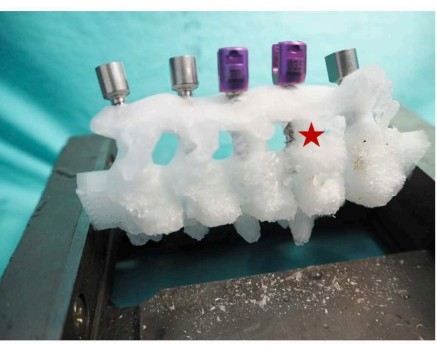

**Fig 6. The pedicle wall perforation was observed from the model after the safety insertion test.** The perforation grade 1 (yellow star) and grade 2 (red star) were observed in standard CPS insertion.

areas of pedicle perforation were observed at the inferior and lateral walls. The proportion of wall perforation was significantly higher in CPS screw insertion compared to the novel CPS insertion 26.7% VS 0%, $P < 0.02$, respectively (Table 2). The experiment showed that the new CPS design had less pedicle wall perforation compared to the conventional CPS design.

### 3. The pullout strength test

A total of 14 screws were tested, including 5 novel CPS screws, 5 conventional CPS screws, and 4 lateral mass screws. One lateral mass screw was not secured to the foam model after applying the pulling jig and was excluded from the experiment. The mean pullout strength, in newtons (N), for each experimental group was illustrated in Fig 7. The highest pullout strength was observed in the standard CPS ($495.01 \pm 123.29$ N), comparable to the new CPS ($449.65 \pm 49.00$ N). The lowest pullout strength was observed in the lateral mass screw ($168.32 \pm 47.79$). The mean difference of the pullout strength between the screws group was illustrated in Table 3. No statistically significant difference in the pullout strength was demonstrated between the new CPS and the standard CPS screw. According to the pull-out strength analysis, the novel CPS screw design yielded a pullout strength comparable to the conventional CPS screw. The pullout strength of the LMS screw was inferior to that of the novel CPS screw.

## Discussion

Our study has highlighted a new design of cervical pedicle screw. With finite element studies, we have proposed a new screw design, with tapered pitch configuration, allowing for non-inferior strength when compared with conventional cervical pedicle screws and superior pullout strength relatively to that of lateral mass screw fixation. Insertion of the novel design screw on 3D printed model has shown lower rate of pedicle perforation, ensuring its safety for use.

**Table 2. Proportion of pedicle wall perforation between the 2 types of screws.**

| Type of screws | Total (N) | No perforation | Pedicle wall perforation | | Percent perforation | P value |
| --- | --- | --- | --- | --- | --- | --- |
| | | | Grade 1 | Grade 2 | | |
| New CPS | 30 | 30 | 0 | 0 | 0 | 0.02 |
| Standard CPS | 30 | 22 | 6 | 2 | 26.7% | |

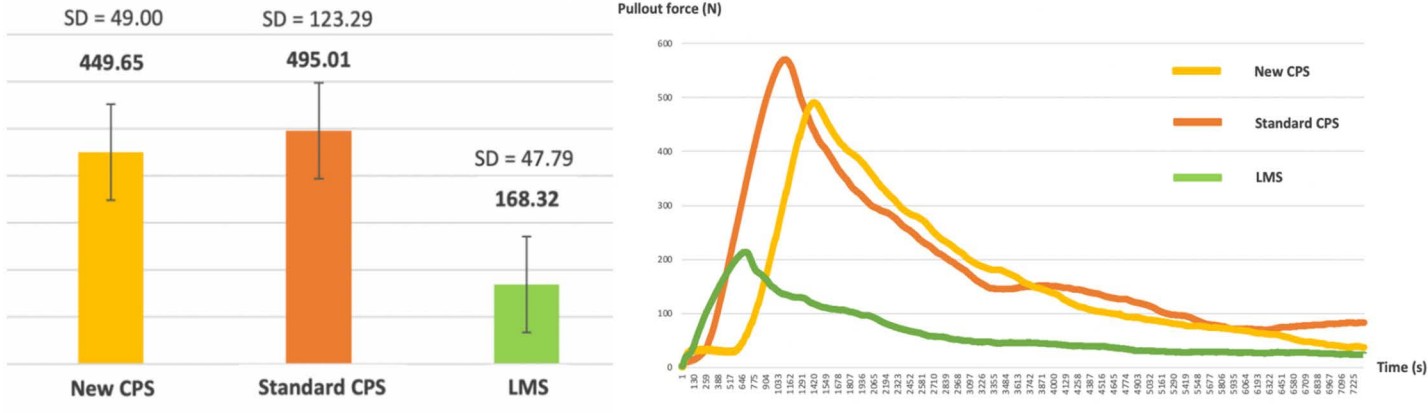

**Fig 7. Pullout strength comparison of three posterior cervical fixation methods.**

**Table 3. Mean difference and P-value compared in each group.**

| Type of screw | Pullout strength (N) | Mean difference | | |
|---|---|---|---|---|
| | | Standard CPS | New CPS | LMS |
| Standard CPS | 495.01 (123.3) | – | 45.35[t] | 326.69* |
| New CPS | 449.65 (49.0) | −45.35[t] | – | 281.33** |
| LMS | 168.32 (47.8) | −281.33** | −326.69* | – |

The DATA showed by mean (SD), * significant with P value = 0.001 **; significance with P value < 0.001; [t] No statistically significant.

Of the numerous techniques for stabilizing the cervical spine, transpedicular screw fixation provides the greatest stability. However, the method of fixation is still technically demanding as it carries risk of catastrophic damages to the surrounding neurovascular structures. Several novel technologies have been proposed to improved CPS insertion accuracy and lessen its associated complications. The using of O-arm based navigation system could reduce the risk of pedicle wall penetration in cervical pedicle screw (CPS) insertion [10]. Wada et al. reported that the pedicle injury was found by 1.2% to 3.8% based on the equipment that used in the surgery and most of the pedicle wall penetration was all less than 2 mm [11]. Using intraoperative computed tomography guided navigation, Blume et al. reported the minimally invasive technique for CPS insertion [12]. Nevertheless, the study has only shown 80% accuracy of insertion and 10% rate of pedicle perforation [12]. Marengo et al. proposed the use of customized patient-specific 3D-printed templates to improve the accuracy and safety of CPS insertion [13]. Amongst 115 CPS inserted, the accuracy of insertion was 93.1% and rate of pedicle breach was 6.9% [13]. Other novel technologies proposed were involved using robotic arm and augmented reality head mounts [14,15]. With the robotic arm, Yamamoto et al. had reported in insertion accuracy to be 97.1% and 97.7% in sagittal and axial planes, respectively [14]. The mean screw deviation was less 1 mm at any point, without any instrument related complications [14]. Ruiz-Cardozo et al. the use of augmented reality assisted insertion of CPS, achieving 100% accuracy upon 41 screw insertions without any significant pedicle breach [15].

Nevertheless, despite the cutting-edge technology, the accuracy of pedicle screw insertion was relatively low especially in mid cervical level [16]. In a systematic review of over 14118 CPS insertions, Irmak et al. reported the accuracy of sub-axial CPS insertion to be around 84%, with a 1.2% perioperative complication rate attributed to screw malposition [17].

Our previous study illustrated that cervical pedicle diameter is relatively small and may not be a good match for the conventional 3.0- and 3.5-mm screws currently available for use as cervical pedicle screws [6]. This could result in pedicle wall violation, which has been highlighted in many studies [10,11,18].

In this study, the novel cervical pedicle screw was designed to overcome the limitation of the small cervical pedicle. Our screw design was a tapered configuration with a smaller core and thread diameter at the middle and lower part of the screw, which could help mitigate the pedicle wall perforation. With a smaller screw diameter, the pullout strength would be significantly reduced compared to the conventional screw design. Thus, our novel screw design increases the thread diameter at the upper part of the screw to overcome this downfall (Fig 1 and 7).

Two models of novel CPS designs were proposed in the finite element study. In CPS model 1, the screws are composed of a tapered shape with a constant pitch. This model was tested and revealed inferior pullout strength compared to the standard CPS. Thus, in our CPS model 2, the screws were redesigned with a decreased pitch length at the proximal part of the screw. This helped improve the bony purchase at the facet joint level. The benefit of this design is affirmed by the comparable pullout strength to that of the standard CPS.

In the phase 2 study, the safety of screw instrumentation was tested in the 3D-printed models of the human cervical spine. Lamina and spinous process were subtracted from the model to allow for the detection, under direct vision, of pedicle perforation upon screw insertions. This also permits better visualization of the pedicle axis, and inner pedicle anatomy to determine the suitable trajectory screw insertion. Furthermore, the screw pilot hole was created via computer simulation before printing the models. This mimics screw insertion under navigation in a real-life setting, enhancing the insertion accuracy.

In the novel CPS group, a significantly lower rate of pedicle wall perforation is observed compared to that in the conventional CPS (p = 0.002). Owing to the taper configuration, the novel design CPS had a relatively smaller thread diameter at the thread-shaft junction at the intra-pedicular portion of the screw. This helped decrease the chance of pedicle wall breakage in this small area. Additionally, the small thread diameter and self-tapping screw head give the advantage of less bony violation when a suboptimal screw trajectory is initially obtained. In the standard CPS group, the higher rate of pedicle wall perforation is observed due to the relatively large thread diameter at the intra-pedicular portion of the screw when compared with the pedicle dimension as observed in previous literatures [11,16]. Although the aiming tracts were pre-designed from CT images to ensure optimal screw trajectories, pedicle wall perforations still occurred in the conventional CPS group. Given that all insertions were performed by a single experienced spine surgeon using a standardized protocol, these perforations likely reflect the design-related mismatch between the screw dimensions and the pedicle anatomy, rather than technical insertion error. This highlights the importance of tailored screw design in improving insertion safety.

A biomechanical study on tri-layer and mono-layer foam models with a density comparable to that of the cervical pedicle and lateral mass was performed. In the pullout strength test, the novel design cervical pedicle screw was shown to have a relatively lower pullout strength compared to the conventional CPS but did not demonstrate statically significance (p = 1.00). The actual pullout forces were 495.01 ± 123.29 N in conventional CPS and 449.65 ± 49 N in the new CPS model. Considering the effect of screw shape, conical screws have been shown to provide better pullout strength compared with cylindrical screws in lumbar pedicle screw fixation [7,8]. However, this concept may not be applicable in cervical pedicle screw fixation as the cervical spine has very short pedicles relative to the lumbar spine. However, this finding needs to be reconfirmed using an anatomical model or a cadaveric study.

Although the tapered screw design may allow for safer instrumentation – showing less pedicle perforation, the smaller screw overall diameter should lead to a less bony purchase and, hence, decrease pullout strength. Our finite element model highlighted that the initial fixation strength is enhanced by modifying the pitch and thread diameter at the cortico-cancellous bone junction. This led to our final screw design, which eventually allowed for an improved pullout strength.

Amongst the tested screw designs, the lateral mass screw had the lowest pullout force – a strength significantly lower than that of the novel design CPS (p = 0.001) and conventional CPS (p < 0.001). The inferiority of pullout strength should be owed to the shorter screw length when compared with CPS. Furthermore, lateral mass screw fixation only traverses a

channel of cancellous bone, as represented in the mono-layer foam block. Thus, the amount of cortical bone purchased is much less in lateral mass screws compared with cervical pedicle screws.

Our study is not without limitations. Firstly, we were unable to conduct the safety and biomechanics tests in the human cadaver due to COVID-19 pandemic during the period of our study. Thus, the study was conducted in the 3D printed cervical spine models and sawbones instead. Secondly, the sample size used in our study was relatively small. Thirdly, our finite element analysis focused solely on evaluating axial pullout strength to optimize screw thread and core design. However, the stress and strain distributions within the screw and surrounding bone—particularly Von Mises stress—were not analyzed. For future studies, larger scale study conducted in human cadavers should be performed ensure the result of this study. Tests in cadaveric samples should allow for more accurate determination of safety of insertion and pullout strength. Future studies incorporating finite element analysis under simulated physiological loads (e.g., flexion and extension) with full posterior construct fixation using rods and links will be essential to fully understand the mechanical behavior of the novel screw in a dynamic, clinically relevant environment. If the enhanced safety and non-inferior fixation strength can be confirmed, our new screw design should provide a strong alternative method for posterior cervical fixation.

## Supporting information

**S1 Data. Raw Data. S2 Data.** Human Participants Research Checklist.
(XLSX)

## Author contributions

**Conceptualization:** Pilan Jaipanya, Pongsthorn Chanplakorn, Annop Sikongkaew, Anak Khantachawan, Suphaneewan Jaovisidha, Thamrong Lertudomphonwanit.

**Data curation:** Pilan Jaipanya, Pongsthorn Chanplakorn, Annop Sikongkaew, Suphaneewan Jaovisidha, Thamrong Lertudomphonwanit.

**Formal analysis:** Pilan Jaipanya, Pongsthorn Chanplakorn, Annop Sikongkaew, Suphaneewan Jaovisidha, Thamrong Lertudomphonwanit.

**Funding acquisition:** Pongsthorn Chanplakorn, Annop Sikongkaew, Anak Khantachawan, Suphaneewan Jaovisidha, Thamrong Lertudomphonwanit.

**Investigation:** Pilan Jaipanya, Pongsthorn Chanplakorn, Annop Sikongkaew, Anak Khantachawan, Suphaneewan Jaovisidha, Thamrong Lertudomphonwanit.

**Methodology:** Pongsthorn Chanplakorn, Annop Sikongkaew.

**Project administration:** Pongsthorn Chanplakorn, Annop Sikongkaew, Anak Khantachawan, Thamrong Lertudomphonwanit.

**Resources:** Pongsthorn Chanplakorn, Annop Sikongkaew, Anak Khantachawan, Thamrong Lertudomphonwanit.

**Software:** Pongsthorn Chanplakorn, Annop Sikongkaew.

**Supervision:** Pongsthorn Chanplakorn, Annop Sikongkaew, Suphaneewan Jaovisidha.

**Validation:** Pilan Jaipanya, Pongsthorn Chanplakorn, Annop Sikongkaew, Anak Khantachawan, Suphaneewan Jaovisidha, Thamrong Lertudomphonwanit.

**Visualization:** Pilan Jaipanya, Pongsthorn Chanplakorn, Annop Sikongkaew, Suphaneewan Jaovisidha, Thamrong Lertudomphonwanit.

**Writing – original draft:** Pilan Jaipanya, Pongsthorn Chanplakorn, Annop Sikongkaew.

**Writing – review & editing:** Pilan Jaipanya, Pongsthorn Chanplakorn, Annop Sikongkaew, Thamrong Lertudomphonwanit.

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
