## [Decision Letter · Decision Letter 0]

13 May 2025

PONE-D-25-13101Novel cervical pedicle screw design to enhance the safety insertion without compromising biomechanical strengthPLOS ONE

Dear Dr. Chanplakorn,

Thank you for submitting your manuscript to PLOS ONE. After careful consideration, we feel that it has merit but does not fully meet PLOS ONE’s publication criteria as it currently stands. Therefore, we invite you to submit a revised version of the manuscript that addresses the points raised during the review process.

We look forward to receiving your revised manuscript.

Kind regards,

KOICHIRO ONO

Academic Editor

PLOS ONE

Journal Requirements:

3. Thank you for stating the following financial disclosure: [The research is funded the Faculty of Medicine, Ramathibodi Hospital, Mahidol University, Bangkok, Thailand].  

Reviewers' comments:

Reviewer's Responses to Questions

**Comments to the Author**

1. Is the manuscript technically sound, and do the data support the conclusions?

Reviewer #1: Yes

Reviewer #2: Partly

2. Has the statistical analysis been performed appropriately and rigorously? 

Reviewer #1: Yes

Reviewer #2: I Don't Know

3. Have the authors made all data underlying the findings in their manuscript fully available?

Reviewer #1: Yes

Reviewer #2: Yes

4. Is the manuscript presented in an intelligible fashion and written in standard English?

Reviewer #1: Yes

Reviewer #2: No

5. Review Comments to the Author

Reviewer #1: Review

Title

Novel cervical pedicle screw design to enhance the safety insertion without compromising biomechanical strength the objectives and results reflect the title of the article

Abstract

1. The authors refer to a previous study. It is necessary to specify the previous study (line40)

2. The results do not include figures reflecting strength. The authors also provide comments and interpretations of the results.

Introduction

1. The current created Cervical transpedicular screw fixators which are recommended by international guidelines and societies are not disclosed. If such are absent, the authors should indicate the absence or presence

Methods

1. Authors presented clearly and detailed information on methodology chapter

2. Safety, pullout strength, and screw insertion technique presented in the chapter

3. authors provided data on Ethics approval (MURA2020/115).

Results

1. Data performed chronologically

2. Data on chapter and in figures/tables equal

3. There is no interpretation of the results

Discussion

1. Chapter is not started from concise statement summarizing the main findings of the study : “Of the numerous techniques for stabilizing the cervical spine, transpedicular screw fixation 244 provides the greatest stability. However, the fixation method is still technically demanding as it carries 245 the risk of catastrophic damage to the surrounding neurovascular structures.”

2. data from similar or previous studies provided.

3. the chapter contains repetitions of information from the results chapter

4. Authors presented limitations of the study

5. There is no highlighted statement of the study for future direction

References

Limited number of used references - 13.

Line 346 – 22?

84% of all references provided are outdated (more than 10 years)

Decision

Minor revision

Reviewer #2: 1. In the section on screw insertion technique, the conventional CPS insertion method involved predrilling with a 2.05-mm drill, followed by tapping with a 3.5-mm tap before screw insertion. In contrast, the novel technique utilized a 2.00-mm drill. The feasibility of maintaining a precise 2.05-mm drill diameter is questionable. Additionally, is the 0.05-mm difference a significant factor influencing outcomes?

2. The inclusion of Von Mises stress and strain analysis for both screws and vertebrae should be considered, as the current data only address pullout strength. Additionally, assessing stress and strain under flexion and extension with additional fixation using rods and links would also be of interest.

3. For each pedicle, an individualized aiming tract was created on CT scan images to ensure optimal screw insertion trajectories prior to model printing, using image processing software. Despite this pre-planning, some screw perforations were still observed. Could this be attributed to technical errors during the insertion process?

4. The novel screw shown in Figures 1 and 6 appears to be monoaxial, whereas the manuscript describes the conventional screws as multiaxial. Is there a difference in pullout strength between monoaxial and multiaxial screws?

5. LMS was also used for comparison; however, detailed information regarding the depth of screw insertion was not provided. Could the data be presented for the four groups as follows: TPS with conventional screws, TPS with novel screws, LMS with conventional screws, and LMS with novel screws? Additionally, the use of LMS techniques appears to be more common than TPS.

6. PLOS authors have the option to publish the peer review history of their article (what does this mean? ). If published, this will include your full peer review and any attached files.

**Do you want your identity to be public for this peer review?** For information about this choice, including consent withdrawal, please see our Privacy Policy .

Reviewer #1: No

Reviewer #2: No

---

## [Author Response · Author response to Decision Letter 1]

23 Jun 2025

Responses to reviewer

1. Is the manuscript technically sound, and do the data support the conclusions?

Reviewer #1: Yes

Authors’ responses: The authors would like the thanks the reviewer for considering our manuscript as a technically sound piece of scientific research.

Reviewer #2: Partly

Authors’ responses: The authors would like the thanks the reviewer for considering our manuscript as a technically sound piece of scientific research. The authors wishes to improve the quality of the manuscipt with this revision based of the reviewer’s expert comments.

2. Has the statistical analysis been performed appropriately and rigorously?

Reviewer #1: Yes

Authors’ responses: The authors would like the thanks the reviewer for considering statistical analysis.

Reviewer #2: I Don't Know

Authors’ responses: The authors would like the thanks the reviewer for considering statistical analysis. We wish to display more clearly in the revised manuscript that the statistical analysis have been appropriately and rigorously performed.

3. Have the authors made all data underlying the findings in their manuscript fully available?

Reviewer #1: Yes

Authors’ responses: The authors would like the thanks the reviewer for considering our shared data.

Reviewer #2: Yes

Authors’ responses: The authors would like the thanks the reviewer for considering our shared data.

4. Is the manuscript presented in an intelligible fashion and written in standard English?

Reviewer #1: Yes

Authors’ responses: The authors would like the thanks the reviewer for approving the language of the submitted article.

Reviewer #2: No

Authors’ responses: The authors would like the thanks the reviewer for considering the language of the submitted article. Authors will improve the written language of the manuscript to be more clear, correct, and unambiguous.

5. Review Comments to the Author

Reviewer #1: Review

Title

Novel cervical pedicle screw design to enhance the safety insertion without compromising biomechanical strength the objectives and results reflect the title of the article

Authors’ responses: The authors would like the thanks the reviewer for approving the title of our manuscript.

Abstract

1. The authors refer to a previous study. It is necessary to specify the previous study (line40)

2. The results do not include figures reflecting strength. The authors also provide comments and interpretations of the results.

Authors’ responses: The authors would like the thanks the reviewer for considering the abstract section of our manuscript.

1. We have mentioned the reference to our previous study in the revised abstract (line 40 – 41)

2. We have included the details on pullout strength for better understanding of our result (line 49-52)

Introduction

1. The current created Cervical transpedicular screw fixators which are recommended by international guidelines and societies are not disclosed. If such are absent, the authors should indicate the absence or presence

Authors’ responses: The authors would like the thanks the reviewer this informative comment. We have added that there is currently a lack of guideline on the standard choice of posterior cervical fixation as highlighted in line 58, 65- 67.

Methods

1. Authors presented clearly and detailed information on methodology chapter

2. Safety, pullout strength, and screw insertion technique presented in the chapter

3. authors provided data on Ethics approval (MURA2020/115).

Authors’ responses: The authors would like the thanks the reviewer for approving the methodology section of our manuscript.

Results

1. Data performed chronologically

2. Data on chapter and in figures/tables equal

3. There is no interpretation of the results

Authors’ responses: The authors would like the thanks the reviewer for approving the result section of our manuscript. We have added a more on the interpretation of our result for easier understanding. The change in be seen in line 254-255 and line 286 – 290.

Discussion

1. Chapter is not started from concise statement summarizing the main findings of the study : “Of the numerous techniques for stabilizing the cervical spine, transpedicular screw fixation 244 provides the greatest stability. However, the fixation method is still technically demanding as it carries 245 the risk of catastrophic damage to the surrounding neurovascular structures.”

2. data from similar or previous studies provided.

3. the chapter contains repetitions of information from the results chapter

4. Authors presented limitations of the study

5. There is no highlighted statement of the study for future direction

Authors’ responses: The authors would like the thanks the reviewer for considering our discussion section.

1. The authors have add a concise statement summaring the main findings of the study in line 322 – 326.

2. The authors have provided data from previous studies and have discussed how our study have contributed to buidling new knowledge.

3. Some repetition of information from the result section were essential to provide clarity upon discussion of our results with previous studies.

4. We have stated our limitations and would hope for better study design in future studies.

5. The authors have added a statement in line 434 - 433 regarding the future direction of any subsequent studies and the use of our newly designed cervical pedicle screw.

References

Limited number of used references - 13.

Line 346 – 22?

84% of all references provided are outdated (more than 10 years)

Authors’ responses: The authors would like the thanks the reviewer for considering the reference literatures that were cited in our manuscripts. The authors have added severeal updates on the current technologies that were recently proposed to help tackle with the low accuracy of CPS insertion and their associated complications. The updated discussion based of new literature can be finded in lines 322-443. The authors have cited five new literatures with relavant findings which are published in the last 3 years.

Reviewer #2:

1. In the section on screw insertion technique, the conventional CPS insertion method involved predrilling with a 2.05-mm drill, followed by tapping with a 3.5-mm tap before screw insertion. In contrast, the novel technique utilized a 2.00-mm drill. The feasibility of maintaining a precise 2.05-mm drill diameter is questionable. Additionally, is the 0.05-mm difference a significant factor influencing outcomes?

Authors’ responses:

The authors appreciate the reviewer’s attention to this technical point. The conventional screw protocol followed the manufacturer’s recommendation of 2.05-mm predrilling and 3.5-mm tapping. The novel screw, designed with a self-tapping tip, was inserted after a 2.00-mm pilot hole without tapping. The choice of 2.00 mm was based on optimizing thread engagement given the self-tapping nature of the screw. While the 0.05-mm difference is minor and within standard manufacturing tolerances, we agree that it alone is unlikely to cause a significant difference in pullout strength. The more critical distinction is the omission of the tapping step in the novel screw group, which may alter the screw–bone interface. We have clarified this point in the revised manuscript; line 214 - 221.

2. The inclusion of Von Mises stress and strain analysis for both screws and vertebrae should be considered, as the current data only address pullout strength. Additionally, assessing stress and strain under flexion and extension with additional fixation using rods and links would also be of interest.

Authors’ responses:

The authors would like to thank the reviewer for this valuable and constructive suggestion. In our current study, the primary focus was to evaluate the biomechanical pullout strength and safety of pedicle wall integrity based on the initial screw–bone interface, without the application of external loads. The finite element analysis (FEA) performed was limited to axial pullout force modeling to optimize the tapering design of the novel screw.

We agree that incorporating Von Mises stress and strain distribution for both screws and surrounding bone would provide further insight into local mechanical behavior, especially under physiological loading conditions such as flexion, extension, and combined motion with rod fixation. These additional analyses would be ideal in the next phase of development and validation of the novel screw, particularly in multilevel fixation scenarios.

We have included this point in the revised discussion section as a limitation and a direction for future research as noted in line 431 - 443.

3. For each pedicle, an individualized aiming tract was created on CT scan images to ensure optimal screw insertion trajectories prior to model printing, using image processing software. Despite this pre-planning, some screw perforations were still observed. Could this be attributed to technical errors during the insertion process?

Authors’ responses:

The authors appreciate the reviewer’s thoughtful question. The individualized aiming tracts were indeed created based on CT imaging to guide ideal screw trajectories during 3D model printing. However, despite this controlled pre-planning, pedicle wall perforations were observed in the conventional screw group.

We believe that the perforations are not solely attributable to technical error, as the same surgeon, using the same insertion protocol, performed all screw insertions. Rather, the perforations may reflect the limitations of the conventional screw design, especially its larger and uniform thread diameter, which may exert lateral pressure against the pedicle wall—particularly in narrow or anatomically variable pedicles—even when the trajectory is optimal. In contrast, the novel screw’s tapered configuration allowed better accommodation within the pedicle, contributing to its perforation-free insertion performance under the same conditions.

We have clarified this point in the discussion section to emphasize that perforation in the standard CPS group likely results from the interaction between screw geometry and pedicle morphology, rather than insertion technique alone. The change is noted in line 403 – 408.

4. The novel screw shown in Figures 1 and 6 appears to be monoaxial, whereas the manuscript describes the conventional screws as multiaxial. Is there a difference in pullout strength between monoaxial and multiaxial screws?

Authors’ responses:

The authors thank the reviewer for pointing this out. Indeed, the conventional screw used was a polyaxial design, while the novel screw was monoaxial. However, in our biomechanical pullout test, the pullout force was applied axially along the screw shaft using a custom jig that directly attached to the screw shank—not the tulip or head portion. As such, the articulation of the tulip in polyaxial screws had no mechanical role during the test.

Previous studies have demonstrated that pullout strength is primarily influenced by the screw–bone interface geometry, such as thread design, diameter, and pitch, rather than by the head design (monoaxial vs. polyaxial), particularly when force is applied directly to the shaft.

We have clarified this point in the Materials and Methods section; line 201 – 205.

5. LMS was also used for comparison; however, detailed information regarding the depth of screw insertion was not provided. Could the data be presented for the four groups as follows: TPS with conventional screws, TPS with novel screws, LMS with conventional screws, and LMS with novel screws? Additionally, the use of LMS techniques appears to be more common than TPS.

Authors’ responses:

We appreciate the reviewer’s thoughtful observation. In our study:

• The conventional cervical pedicle screws (TPS) and novel screws were inserted into tri-layer foam blocks to simulate pedicle screw trajectories.

• The lateral mass screws (LMS) were inserted into mono-layer cancellous foam blocks, representing lateral mass anatomy.

Regarding screw types:

• Only one type of screw was used per fixation type:

o TPS with conventional CPS

o TPS with novel CPS

o LMS with conventional LMS screws

o LMS with novel screws was not tested, as the novel screw design was specifically intended for pedicle fixation and not optimized for lateral mass use.

Regarding screw depth:

• The TPS (both conventional and novel) screws were 22 mm in length, fully inserted into tri-layer blocks (10 mm cortical, 10 mm cancellous, 10 mm cortical).

• The LMS screws were 16 mm in length, fully inserted into mono-layer cancellous blocks of 30 mm thickness.

We have revised the methods to clarify insertion depth, what screw lengths were used, and the reason in which the novel CPS was not used for LMS insertion. The change is noted in line 192 – 197.

---

## [Decision Letter · Decision Letter 1]

10 Aug 2025

Novel cervical pedicle screw design to enhance the safety insertion without compromising biomechanical strength

PONE-D-25-13101R1

Dear Dr. Chanplakorn,

We’re pleased to inform you that your manuscript has been judged scientifically suitable for publication and will be formally accepted for publication once it meets all outstanding technical requirements.

Kind regards,

KOICHIRO ONO

Academic Editor

PLOS ONE

Additional Editor Comments (optional):

Reviewers' comments:

Reviewer's Responses to Questions

**Comments to the Author**

1. If the authors have adequately addressed your comments raised in a previous round of review and you feel that this manuscript is now acceptable for publication, you may indicate that here to bypass the “Comments to the Author” section, enter your conflict of interest statement in the “Confidential to Editor” section, and submit your "Accept" recommendation.

Reviewer #1: All comments have been addressed

2. Is the manuscript technically sound, and do the data support the conclusions?

Reviewer #1: Yes

3. Has the statistical analysis been performed appropriately and rigorously? 

Reviewer #1: Yes

4. Have the authors made all data underlying the findings in their manuscript fully available?

Reviewer #1: Yes

5. Is the manuscript presented in an intelligible fashion and written in standard English?

Reviewer #1: Yes

6. Review Comments to the Author

Reviewer #1: authors provided answers according all comments

My comments were advisory in nature

7. PLOS authors have the option to publish the peer review history of their article (what does this mean? ). If published, this will include your full peer review and any attached files.

**Do you want your identity to be public for this peer review?** For information about this choice, including consent withdrawal, please see our Privacy Policy .

Reviewer #1: **Yes: ** Raikhan Bolatbekova

---

## [Editor Report · Acceptance letter]

PONE-D-25-13101R1

PLOS ONE

Dear Dr. Chanplakorn,

I'm pleased to inform you that your manuscript has been deemed suitable for publication in PLOS ONE. Congratulations! Your manuscript is now being handed over to our production team.

Kind regards,

on behalf of

Dr. KOICHIRO ONO

Academic Editor

PLOS ONE